# Phytoextraction and Migration Patterns of Cadmium in Contaminated Soils by *Pennisetum hybridum*

**DOI:** 10.3390/plants12122321

**Published:** 2023-06-15

**Authors:** Canming Chen, Zebin Wei, Kuangzheng Hu, Qi-Tang Wu

**Affiliations:** Guangdong Provincial Key Laboratory of Agricultural & Rural Pollution Abatement and Environmental Safety, College of Natural Resources and Environment, South China Agricultural University, Guangzhou 510642, China; chencanming@stu.scau.edu.cn (C.C.); wezebin@scau.edu.cn (Z.W.); hu15814516036@163.com (K.H.)

**Keywords:** *Pennisetum hybridum*, cadmium, phytoextraction, migration, removal pathways

## Abstract

This study was conducted to identify soil cadmium (Cd) removal pathways and their contribution rates during phytoremediation by *Pennisetum hybridum*, as well as to comprehensively assess its phytoremediation potential. Multilayered soil column tests and farmland-simulating lysimeter tests were conducted to investigate the Cd phytoextraction and migration patterns in topsoil and subsoil simultaneously. The aboveground annual yield of *P. hybridum* grown in the lysimeter was 206 ton·ha^−1^. The total amount of Cd extracted in *P. hybridum* shoots was 234 g·ha^−1^, which was similar to that of other typical Cd-hyperaccumulating plants such as *Sedum alfredii*. After the test, the topsoil Cd removal rate was 21.50–35.81%, whereas the extraction efficiency in *P. hybridum* shoots was only 4.17–8.53%. These findings indicate that extraction by plant shoots is not the most important contributor to the decrease of Cd in the topsoil. The proportion of Cd retained by the root cell wall was approximately 50% of the total Cd in the root. Based on column test results, *P. hybridum* treatment led to a significant decrease in soil pH and considerably enhanced Cd migration to subsoil and groundwater. *P. hybridum* decreases Cd in the topsoil through multiple pathways and provides a relatively ideal material for phytoremediation of Cd-contaminated acid soils.

## 1. Introduction

Heavy metal pollution has become a global challenge, among which cadmium (Cd) contamination has emerged as one of the most concerning environmental issues [1]. In China, the rate of sites containing Cd in excess of regulatory limits has reached more than 7% [2]. Rice (*Oryza sativa* L.), which is a major grain crop grown worldwide, usually faces a high risk of Cd exposure [3]. Approximately 1/10 of the rice produced in China has a Cd content above the limit of the national food safety standard (0.2 mg·kg^−1^) [4]. As a result, there is an urgent need to control and remediate Cd-contaminated farmland soils.

Phytoextraction is a cost-effective approach that is suitable for the remediation of heavy metal-contaminated soils over large areas and provides high comprehensive benefits [5]. Tolerant plants with high biomass have recently been developed and used for phytoextraction of heavy metals. The hybrid giant napier (*Pennisetum hydridum*) is an energy plant with strong stress resistance, high productivity, and well-developed roots [6]. Based on its tolerance to heavy metals [7], *P. hybridum* has been proposed for phytostabilization of heavy metals in soils [8]. It has also been suggested that *P. hybridum* could be used for phytoextraction of Cd from contaminated soils [9]. Subsequent research showed that ammonium chloride enhanced the removal of soil Cd and Zn by *P. hybridum*, whereas digestate increased its shoot biomass and local Cd accumulation [10]. Additionally, intercropping of *Solanum photeinocarpum* (a hyperaccumulator) with *P. hybridum* led to considerable improvement of the phytoremediation potential for Cd-contaminated soil [11]. However, researchers have often explored the mechanisms by which *P. hybridum* remediates soil through pot experiments with a single soil layer. Moreover, the majority of previous studies have focused on Cd phytoextraction and removal by the shoots of *P. hybridum*, while the indirect effects and phytoremediation potential of the roots have received little attention. There have also been few comprehensive and in-depth analyses of the pathways of Cd removal from soil systems.

The developed roots of *P. hydridum* have a larger bioconcentration factor (BCF) than the shoots, which calls for further investigation of the effects and mechanisms of rhizoremediation. Therefore, multilayered soil column and farmland-simulating lysimeter tests were conducted to investigate the phytoextraction and migration patterns of Cd in different soil layers. The various pathways of Cd removal from soil and their contribution rates during the remediation process were clarified to assess the phytoremediation potential and effects of *P. hydridum.*

## 2. Results

### 2.1. Plant Growth of P. hybridum

Table 1 and Table 2 present the plant growth observed during this study. There was no remarkable stress on plant growth or development of *P. hybridum* in soil under moderate Cd pollution, and normal growth without inhibition at any stages was maintained. The root length of *P. hybridum* reached more than 50 cm under column test conditions, indicating that the developed roots of *P. hybridum* can pass through the topsoil layer (0–20 cm) to enter the subsoil layer. Under the lysimeter test conditions, the plant height of *P. hybridum* was greater than 250 cm in all cases, and the tallest plants reached 312.5 cm. The yield per hectare in each harvest of *P. hybridum* was greater than 4.4 × 10^4^ kg dry weight (DW), which was markedly higher than that of rice grown under the same conditions. These findings verified that *P. hybridum* is a high-biomass-energy plant.

### 2.2. Cd Distribution in Various Parts of P. hybridum

As shown in Figure 1, the distribution of Cd concentrations in various parts of *P. hybridum* followed the order topsoil roots > stems > leaves > subsoil roots (based on column tests; Figure 1a) and topsoil roots > leaves > stems > subsoil roots (based on lysimeter test results; Figure 1c). When compared with other parts, topsoil roots had significantly higher Cd concentrations (α = 0.05), and the weighted mean Cd concentration of roots was higher than that of shoots. The cumulative Cd concentration of *P. hybridum* roots was 2 mg·kg^−1^ under mild pollution and close to 4 mg·kg^−1^ under moderate pollution. These results demonstrate that roots are involved in *P. hybridum* having the highest cumulative concentration of Cd.

In the lysimeter test, the Cd concentrations of *P. hybridum* shoots were significantly higher than those of common Simiao rice and lower than those of upland rice (Figure 1b,c). The shoot Cd concentrations of *P. hybridum* were all less than 2 mg·kg^−1^ dry weight, which corresponded to less than 0.5 mg·kg^−1^ fresh weight based on a 75% water content. These results were below the standard limit (1 mg·kg^−1^) specified in the hygienic standard for feeds (GB 13078-2017), indicating that *P. hybridum* shoots can be used for feed production as well as biofuel.

### 2.3. Cd Uptake and Accumulation Patterns of P. hybridum

#### 2.3.1. Bioconcentration and Translocation Factors of Cd in Various Parts of *P. hybridum*

Since BCF and TF are key reference factors for evaluating a plant’s capacity to take up and translocate heavy metals, they can be used to evaluate the phytoextraction capacity of plants [12]. The BCF and TF values of Cd in various parts of *P. hybridum* are listed in Table 3.

The topsoil roots of *P. hybridum* achieved the highest BCFs for soil Cd (BCF > 2), with larger BCF values being associated with longer planting times. The BCFs varied between 0.65 and 1.25 in the stems and leaves, suggesting that *P. hybridum* is not an accumulator plant. The BCFs in subsoil roots of *P. hybridum* were comparable to those in the leaves. Additionally, higher BCFs were observed for *P. hybridum* in the lysimeter test (close to the field environment), which was most likely a result of the longer time for root growth and Cd accumulation. The TFs in *P. hybridum* shoots were all less than 1, with no significant differences between the stems and leaves.

The above results suggest that the capacity of *P. hybridum* to translocate Cd from roots into shoots is not high, and that *P. hybridum* roots have relatively strong Cd retention.

#### 2.3.2. Location of Cd Storage in *P. hybridum* Roots

We analyzed the Cd subcellular distribution to determine the location and form of Cd stored in *P. hybridum* roots (Figure 2). Of the Cd contained in *P. hybridum* roots, 20.13% was intercepted at the root surface and 49.95% was immobilized in the root cell wall, whereas only 29.92% entered the endoplast. These results indicate that root cell wall binding is the primary mechanism that allows *P. hybridum* to retain heavy metals in its roots.

### 2.4. Phytoextraction Capacity of P. hybridum for Cd

Table 4 presents the amount and efficiency of soil Cd extraction by *P. hybridum* determined in the column test. In test 1, the amount of Cd extracted was highest (0.567 mg·plant^−1^) in topsoil roots of *P. hybridum*. In test 2, the highest amount of Cd extracted (0.1136 mg·plant^−1^) was observed in stems of *P. hybridum*. The Cd extraction efficiency in *P. hybridum* shoots was 4.17% and 5.74% for the two tests, respectively. In test 1, the extraction efficiency was ranked as roots > shoots, whereas in test 2 it was shoots > roots. This may have occurred because the duration of test 2 was longer than that of test 1, allowing the shoot biomass of *P. hybridum* to increase and enhance heavy metals translocation to the shoots. Overall, these results indicate *P. hybridum* can accumulate a larger amount of Cd in its shoots than roots under certain conditions.

We simulated the field environment using percolating filters to further explore the phytoextraction and phytoremediation potential of *P. hybridum* by shoots for soil Cd (Table 5). Four harvests of *P. hybridum* were obtained over one year under the simulated field conditions, with a total Cd extraction amount of 23.40 mg·m^–2^ and a Cd extraction efficiency of 8.53% in the shoots. When upland rice and Simiao rice were grown for one cropping season under the same soil conditions, the total Cd extraction amount in their shoots was only 3.02 mg·m^–2^; however, the shoot Cd extraction of *P. hybridum* was ~7.5 times that of rice. Despite having no prominent ability to translocate Cd to the shoots, *P. hybridum* has a relatively high extraction efficiency for soil Cd because of its high biomass.

### 2.5. Variation of Soil pH and Cd Concentration

#### 2.5.1. Variation of Soil pH

Table 6 and Table 7 show the soil pH under different treatments before and after the test. In both column tests, a significant decrease in the pH of topsoil and subsoil treated with *P. hybridum* occurred. In the lysimeter test, the topsoil and subsoil pH also decreased with increasing harvest under *P. hybridum* treatment. In addition, the pH differed significantly between treatments.

#### 2.5.2. Variation of Soil Cd Concentration

The concentrations of Cd in topsoil treated with *P. hybridum* decreased significantly (α = 0.05) in both the column and lysimeter tests (Figure 3). After 4 months, 6 months, and 1 year of phytoremediation, the topsoil Cd concentrations decreased by 23.62%, 21.50%, and 35.81% in column test 1, column test 2, and the lysimeter test, respectively (Table 8). In the lysimeter test, which was closer to the field environment, the phytoremediation effect on topsoil manifested in the fourth harvest (1 year). The resulting decrease in topsoil Cd concentrations was greater than that observed after planting rice. These findings indicate that *P. hybridum* has excellent phytoremediation effects on Cd-contaminated soils.

Following *P. hybridum* treatment, the Cd concentration of the second soil layer also increased significantly relative to the initial concentration; however, there was no noticeable difference between plants. In the lysimeter test, there was an increase in the Cd concentration of the bottom (third) soil layer, and this increase was more pronounced in the treatment with continuous planting of *P. hybridum*. These results indicate that the roots of *P. hybridum* have the capacity to transfer soil Cd towards lower layers.

### 2.6. Soil Cd Migration and Removal Pathways

We calculated soil Cd migration and removal pathways based on the column test. The distribution patterns of Cd in the soil systems of various treatments after testing are shown in Table 9. The contribution of different pathways to soil Cd removal occurred in the following order: leaching from soil > extraction by shoots > migration to subsoil > translocation in subsoil roots. These results showed that leaching from soil was the largest contributor to topsoil Cd removal, accounting for ~50%. The second largest contributor was extraction by aboveground shoots, which accounted for only 20–30% (Figure 4a).

In the lysimeter test, it was not possible to collect the roots completely. In the topsoil of the lysimeter test, the different removal pathways of Cd were extraction by shoots and migration to subsoil layer 1 (20–40 cm) and migration to subsoil layer 2 (40–90 cm) (sealed at bottom). Overall, the contribution rates of these three removal pathways were 21.37%, 57.49%, and 21.14%, respectively (Figure 4b). Additionally, the rate of extraction by shoots in the lysimeter test was similar to that in the column test. However, migration to subsoil layer 1 made the greatest contribution to soil Cd removal in the lysimeter test, accounting for ~60%. Accordingly, the extent of downward Cd leaching in lysimeters was lower than that in soil columns. Overall, these results indicate that the primary pathway by which *P. hybridum* decreases topsoil Cd is not extraction by shoots, but rather promotion of downward Cd migration.

## 3. Discussion

### 3.1. Advantages of P. hybridum for Phytoremediation of Cd-Contaminated Soils

It has been suggested that *P. hybridum* is highly tolerant to Cd contamination [13]. In the present study, soils contaminated with low concentrations of Cd had little impact on growth of *P. hybridum*, and its annual yield in lysimeters reached 2.06 × 10^5^ kg·ha^−1^ DW. Given the high biomass of *P. hybridum*, it has great potential for removal of Cd from soil [14]. Generally, the effects of plant roots on soil heavy metals is limited to a few millimeters to 1–2 cm from the root surface in the rhizosphere zone [15]. Our observations indicate that the fibrous roots of *P. hybridum* are highly developed and can form fibrous root networks in a relatively short period. Therefore, it is likely that *P. hybridum* roots can extract high levels of heavy metals by contacting more soil particles. Moreover, the developed roots of *P. hybridum* are likely to affect the subsoil below the plow layer. In summary, *P. hybridum* is easy to grow and manage, has a well-developed root system and has high biomass in subtropical areas, which is advantageous for phytoremediation of contaminated soils.

The results of this study showed that the total amount of Cd extracted by *P. hybridum* shoots annually was 234 g·ha^−1^ under simulated field environmental conditions. Among hyperaccumulators grown under field conditions, *Sedum alfredii* Hance can extract 184 g·ha^−1^ of Cd per year, whereas *Sauropus androgynus* (L.) Merr. can only extract 64 g·ha^−1^ per year [16]. Overall, the results of the present study indicate that the amount of Cd extracted by *P. hybridum* shoots can exceed that extracted by *S. alfredii* and *S. androgynus*. Some studies have suggested that the high biomass and rapid growth of *P. hybridum* contribute to a relatively large total extraction amount of heavy metals, despite its poor ability to transfer heavy metals to shoots; accordingly, *P. hybridum* may also be effective at phytoremediation of soils contaminated with heavy metals [17]. Based on a field test, Xie et al. [18] demonstrated that the amount of Cd extracted by *P. hybridum* and the extraction efficiency were 119.9 g·ha^−1^ and 6.98%, respectively, indicating its phytoremediation effect was superior to that of *Solanum nigrum* L., a Cd hyperaccumulator.

After the harvest of the shoots of *P. hybridum*, the Cd contents in the biomass were below the feed standard limit (1 mg/kg) for the tested soils with a Cd content below 1.3 mg/kg soil and can be served as animal feed. This is different to Cd-hyperaccumulating plants whose shoots contain generally too much Cd to be feed and must be handled according to regulations of harmful substances. However, if the soil Cd is higher than 1.3 mg/kg, the biomass should be dried and go to the bioenergy plants to produce electricity [19]. The step-by-step actions necessary for remediating Cd-contaminated soils with *P. hybridum* is proposed in Figure 5.

Based on the decrease in total soil Cd concentration, the rate of Cd removal from the topsoil in column test 1, column test 2, and the lysimeter test was 23.62%, 21.50%, and 35.81%, respectively (Table 8). However, the efficiency of *P. hybridum* extraction by shoots was only 4.17%, 5.74%, and 8.53%, respectively (Table 4 and Table 5), which was significantly different from the soil Cd removal rate. This indicates that extraction by shoots is not the most prominent contributor to phytoremediation of Cd-contaminated soils.

Evaluation of the removal pathways indicated that *P. hybridum* removes more Cd by facilitating its movement from the topsoil to deeper soil layers and groundwater than via extraction by shoots. However, the specific mechanisms by which this occurs were not elucidated. It is possible that these mechanisms are related to leaching being enhanced by the decreased soil pH and root exudates, as the amount of Cd brought directly to the subsoil by plant roots was insignificant as shown in this study.

Ma et al. [20] found that planting with *P. hybridum* considerably neutralized the alkalinity of a contaminated soil and strongly absorbed trace heavy metals such as Zn, Mn, Fe and Cu, causing a significant decrease in the heavy metals concentrations in the contaminated soil. In the present study, a remarkable decrease in soil pH occurred after planting *P. hybridum* (Table 6 and Table 7). The increase in H^+^ concentration that occurred after planting could lead to increased Cd ion exchange at the surface of soil particles, increasing the Cd concentration in soil solution [21]. However, organic acids in root exudates could chelate heavy metal ions and mobilize heavy metals near plant roots [22]. Additional research is required to further decipher the possible dynamic effects of the developed roots and root exudates of *P. hybridum* on the speciation and migration of soil heavy metals.

In situ chemical flushing can cause secondary pollution and damage soil structure. The ‘bioleaching’ of *P. hybridum* roots can be considered an alternative to conventional chemical leaching, which allows Cd removal from the plow layer through downward migration. Previous studies using *P. hybridum* for soil phytoremediation have mainly looked at extraction by shoots, resulting in the phytoremediation efficiency calculated based on the extraction amount in shoots being too conservative and the phytoremediation capacity of *P. hybridum* being underestimated. Improving the phytoremediation capacity of *P. hybridum* necessitates in-depth research of various removal pathways and strengthening the migration-promoting ability of plant roots.

### 3.2. Forms of Cd Stored in Roots of P. hybridum

Root surface interception is a self-protective behavior [23]. The surfaces of rice roots have been found to intercept ~30% of Cd through iron plaques without the addition of exogenous Fe [24]. Zhao et al. [25] reported that early Cd stress response induces transcriptional changes in cell wall remodeling, which may be involved in Cd stress tolerance and Cd ion accumulation in *P. hybridum*. Other similar herbs can effectively detoxify Cr, As, and Pb through cell wall precipitation and vesicular compartmentalization [26]. The results of the present study demonstrate that root cell wall retention is the principal mechanism of Cd storage and detoxification in *P. hybridum* roots, but that root surface interception also plays an important role in this process.

## 4. Materials and Methods

### 4.1. Experimental Materials

Seed stems of *P. hydridum* were provided by the ecological farm of South China Agricultural University (Guangzhou, Guangdong Province, China). All Cd-contaminated soils (topsoils) were collected from actual paddy fields under moderate to mild pollution. The contaminated soil used in column test 1 was obtained from a town in A City, Guangdong Province, while that used in column test 2 and the lysimeter test came from a town in B City, Guangdong Province. Clean soils (subsoils) were collected from the experimental farm of South China Agricultural University (column tests 1 and 2) and a town in B City, Guangdong Province (lysimeter test). The chemical properties and Cd contents of the tested soils are summarized in Table 10.

### 4.2. Experimental Design

#### 4.2.1. Column Test

Time of test 1: 8 September 2020–8 January 2021

Time of test 2: 25 April 2021–25 October 2021

The column test was conducted outdoors with ventilation and light at the experimental farm of South China Agricultural University. In each test, six plexiglass columns with a 10 cm inner diameter and 50 cm height were prepared. Three columns were used as blank controls (no plant), while the other three were planted with one *P. hydridum* each. All columns were wrapped with thin black film and filled with the following materials from bottom to top: 5 cm of quartz sand, 20 cm of uncontaminated soil (subsoil), and 20 cm of Cd-contaminated soil (topsoil). Before starting the test, soils were air-dried and passed through a 1 cm sieve. After the column was filled, it was saturated with tap water. During the test period, the plants in columns were managed with routine practices.

#### 4.2.2. Lysimeter Test (12 July 2021–13 July 2022)

Six lysimeters (square percolating filters) with 1 m long sides were filled with 50 cm of uncontaminated local soil in the bottom layer (the same subsoil used in column test 1 (Table 10)), 20 cm of uncontaminated subsoil in the second layer, and 20 cm of Cd-contaminated topsoil in the surface layer. A black nylon net was spread between each soil layer to separate them. Before beginning the test, the soil was saturated by tap water irrigation. There were two treatments (*P. hydridum* and rice) and three replications, which were randomly assigned to six filters. Rice was grown at a density of 16 plants·m^–2^, with upland rice (cv. Hanyou-73) planted as late rice in 2021 and Simiao rice (cv. Zengcheng) as early rice in 2022. *P. hydridum* was grown at a density of 4 plants·m^–2^. *P. hydridum* was mowed once every three months, after which the stubble was left to re-grow.

### 4.3. Sample Preparation and Analysis

Soil samples were air-dried and ground with an agate mortar, passed through 20- and 100-mesh sieves, and then stored in sealed plastic bags at 0 °C–4 °C prior to analysis. Samples of different plant parts were deactivated in a 70 °C oven for 30 min and then dried at 55 °C until constant weight. Dry plant samples were pulverized using a pulverizer, passed through a 100-mesh sieve, and then stored in sealed plastic bags at 0 °C–4 °C before analysis.

Soil total Cd and plant total Cd were analyzed by microwave digestion-graphite furnace atomic absorption photometry based on the environmental standard method (HJ 832–2017) and the national standard method (GB 5009.15–2014), respectively. Test results were verified using national standard materials for soil (GBW07405a) and plants (GBW(E)100348a). Soil pH was measured with a pH meter at a water:soil ratio of 2.5:1 (*v*/*w*).

One group of fresh root samples of *P. hydridum* was digested with a mixture of concentrated HNO_3_–HClO_4_ (4:1, *v*/*v*) and then used to determine the total Cd content (GB 5009.15-2014). Another group of fresh root samples was soaked in 20 mmol·L^−1^ Na_2_-EDTA to remove the Cd adsorbed at the root surface [27]. After filtration, the Cd concentration in the solution was determined and used to calculate the Cd content at the root surface. Finally, the remaining Cd content in the soaked root samples was determined using the same method for the analysis of root total Cd (GB 5009.15-2014).

The Cd adsorbed by the root surface was removed as previously described [26], after which a methanol–chloroform mixture (2:1, *v*/*v*) was used to remove cell inclusions based on the method described by Hart et al. [28] and used by Sun et al. [29]. This left morphologically intact root cell walls, which were then analyzed for Cd content using the same method that was used for analysis of root total Cd (GB 5009.15-2014).

### 4.4. Data Analysis

Data processing and graphing were accomplished using Microsoft Excel 365 (Microsoft Corp., Redmond, WA, USA). One-way analysis of variance (ANOVA) and Duncan’s multiple range tests were used to identify statistically significant differences between treatments (α = 0.05) using SPSS 21.0 (IBM Corp., Armonk, NY, USA).

The following methods were used to calculate the factors related to heavy metal migration in the soil–plant system:Bioconcentration factor (BCF) = Cd concentration in a plant part/initial soil Cd total concentration;Translocation factor (TF) = Cd concentration in an aerial part/root Cd concentration;Extraction efficiency = Cd extraction amount by plants/initial total Cd content in soil;Removal rate = decrease in soil Cd content after planting/initial total Cd content in soil.

## 5. Conclusions

*P. hybridum* with stubble growth produced high biomass under environmental conditions in lysimeters that simulated field conditions. Specifically, its annual yield reached 206 ton·ha^−1^ DW, and the total amount of Cd extracted was 234 g·ha^−1^ per year.

The total Cd in topsoil decreased significantly following *P. hybridum* treatment, with removal rate of 21.50–35.81% for the tested acid soils. However, the efficiency of Cd extraction by *P. hybridum* shoots was only 4.17–8.53%, indicating that extraction by plant shoots was not the most important contributor to the decrease of Cd in topsoil.

*P. hybridum* treatment resulted in a significant decrease in soil pH. Additionally, *P. hybridum* roots could penetrate the subsoil and enable topsoil Cd to transfer to subsoil layers. Therefore, *P. hybridum* is an ideal material for phytoremediation of Cd-contaminated acid soils in subtropical areas and further investigation of the dynamic effects of its roots on heavy metal speciation and migration are warranted.

## Figures and Tables

**Figure 1 plants-12-02321-f001:**
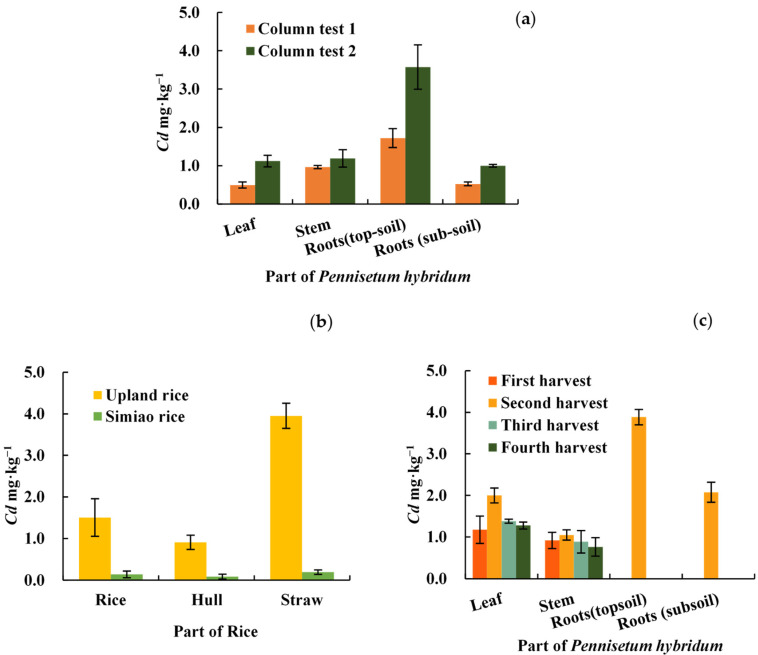
Concentration of Cd in different parts of Pennisetum hybridum and rice (dry weight basis). (**a**) Column test; (**b**,**c**) Lysimeter test.

**Figure 2 plants-12-02321-f002:**
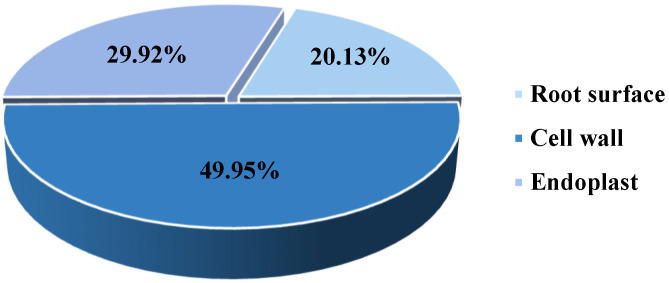
Cd subcellular distribution in roots of *Pennisetum hybridum*.

**Figure 3 plants-12-02321-f003:**
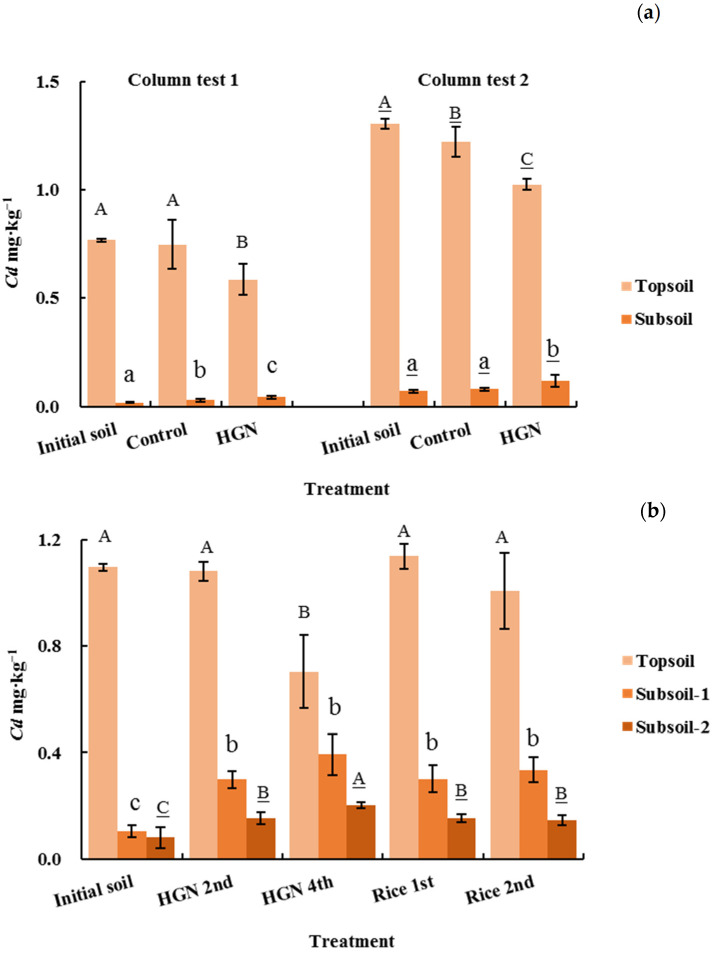
Changes in Cd content in soil before and after the test. (**a**) column tests; (**b**) lysimeter test; HGN: Hybrid giant napier (*Pennisetum hybridum*). Values for the same test and soil affected by the same letter were not significantly different according to Duncan’s multiple range tests (α = 0.05).

**Figure 4 plants-12-02321-f004:**
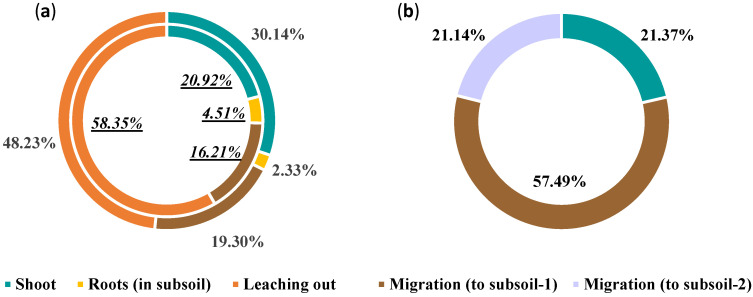
Contribution of different pathways to Cd removal from the topsoil. (**a**) column test (inner ring: test 1, outer ring: test 2); (**b**) lysimeter test.

**Figure 5 plants-12-02321-f005:**
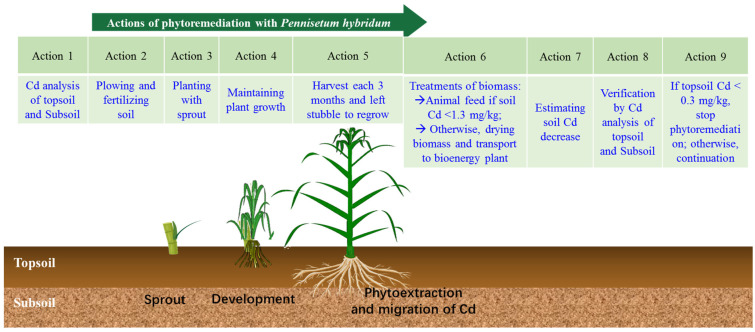
Step-by-step actions for remediating Cd-contaminated soils with *P. hybridum*.

**Table 1 plants-12-02321-t001:** Plant growth and biomass of *Pennisetum hybridum* in column test.

	Root Length (cm)	Height (cm)	Biomass (DW, g·plant^−1^)			
			Root (in Topsoil)	Root (in Subsoil-1)	Stem	Leaf
Test 1	57.67 ± 2.08	174.00 ± 4.58	16.84 ± 1.09	9.60 ± 1.11	44.36 ± 2.52	42.27 ± 7.10
Test 2	51.67 ± 4.51	190.00 ± 8.66	17.91 ± 1.94	11.59 ± 0.90	95.51 ± 2.25	25.96 ± 3.00

**Table 2 plants-12-02321-t002:** Plant growth and biomass in lysimeter test.

	Height (cm)	Biomass (DW, g·plant ^−1^)		Yield (DW, 10^4^ kg·ha^−1^)	
** *Pennisetum hybridum* **		Stem	Leaf	Stem	Leaf
First harvest	312.50 ± 5.45	856.95 ± 92.15	332.86 ± 36.98	3.430	1.332
Second harvest	267.08 ± 12.14	919.83 ± 102.2	555.17 ± 62.00	3.681	2.222
Third harvest	259.17 ± 15.28	625.15 ± 47.32	478.89 ± 36.37	2.502	1.917
Fourth harvest	265.00 ± 10.00	721.52 ± 59.47	653.39 ± 54.68	2.888	2.615
Rice		Straw	Grain	Straw	Grain
Upland rice	130.00 ± 6.00	31.73 ± 1.69	43.29 ± 1.95	0.5080	0.6930
Simiao rice	120.00 ± 5.50	22.96 ± 3.97	15.03 ± 1.48	0.3675	0.2406

**Table 3 plants-12-02321-t003:** Bioconcentration factors (BCF) and translocation factors (TF) of Cd for different parts of *Pennisetum hybridum*.

	BCF				TF	
	Root (in Topsoil)	Root (in Subsoil-1)	Stem	Leaf	Stem	Leaf
Column test 1	2.24	0.68	1.25	0.65	0.56	0.29
Column test 2	2.74	0.76	0.91	0.86	0.33	0.31
Lysimeter test	3.54	1.89	0.95	1.82	0.27	0.51

**Table 4 plants-12-02321-t004:** Extraction amount and efficiency of soil Cd by *Pennisetum hybridum* in column test.

	Extraction Amount (mg·plant^−1^)					Extraction Efficiency (%)	
	Root (in Topsoil)	Root (in Subsoil-1)	Stem	Leaf	Total	Roots	Shoots
Test 1	0.0567 ± 0.0044	0.0138 ± 0.0021	0.0428 ± 0.0043	0.0212 ± 0.0060	0.1345	4.60	4.17
Test 2	0.0642 ± 0.0086	0.0116 ± 0.0008	0.1136 ± 0.0022	0.0363 ± 0.0034	0.2256	2.90	5.74

**Table 5 plants-12-02321-t005:** Amount and efficiency of Cd extraction by *Pennisetum hybridum* or rice in lysimeter test.

	Extraction Amount (mg·m^–2^)				Extraction Efficiency (%)
*Pennisetum hybridum*	Stem	Leaf	Each Harvest	Annual	
First harvest	3.153 ± 0.085	1.564 ± 0.042	4.717	23.40	8.53
Second harvest	3.850 ± 0.107	4.438 ± 0.123	8.288		
Third harvest	2.216 ± 0.041	2.641 ± 0.049	4.857		
Fourth harvest	2.207 ± 0.046	3.334 ± 0.069	5.541		
**Rice**	**Straw**	**Grain**			
Upland rice	2.006 ± 0.107	0.913 ± 0.041	2.919	3.02	1.10
Simiao rice	0.071 ± 0.012	0.029 ± 0.003	0.100		

**Table 6 plants-12-02321-t006:** Variation of soil pH after planting *Pennisetum hybridum* in column test.

	Test 1			Test 2		
	Initial Value	Control	*P. hybridum*	Initial Value	Control	*P. hybridum*
Topsoil	5.57 ± 0.03 A	5.20 ± 0.27 B	4.87 ± 0.05 C	5.27 ± 0.01 a	5.14 ± 0.01 b	4.81 ± 0.01 b
Subsoil-1	4.47 ± 0.03 A	5.20 ± 0.28 A	3.76 ± 0.12 B	6.27 ± 0.04 a	6.42 ± 0.02 a	5.86 ± 0.19 b

Means in the same row followed by the same letter were not significantly different according to Duncan’s multiple range tests (α = 0.05).

**Table 7 plants-12-02321-t007:** Variation of soil pH after planting in lysimeter test.

	Initial Value	*Pennisetum hybridum*		Rice	
		Second Harvest	Fourth Harvest	Upland Rice	Simiao Rice
Topsoil	5.58 ± 0.11 B	5.06 ± 0.14 C	4.37 ± 0.23 D	5.47 ± 0.10 B	5.95 ± 0.31 A
Subsoil-1	5.59 ± 0.25 A	5.10 ± 0.03 C	4.76 ± 0.17 D	5.45 ± 0.03 AB	5.30 ± 0.14 BC

Means in the same row followed by the same letter were not significantly different according to Duncan’s multiple range tests (α = 0.05).

**Table 8 plants-12-02321-t008:** Topsoil Cd concentration and removal rate.

	Cd Concentration (mg·kg^−1^)		Cd Removal Rate (%)
	Initial	After Planting	
Column test 1	0.7676 ± 0.0077	0.5863 ± 0.0429	23.62
Column test 2	1.3058 ± 0.0232	1.0250 ± 0.0265	21.50
Lysimeter test	1.0970 ± 0.0123	0.7041 ± 0.1385	35.81

**Table 9 plants-12-02321-t009:** Cd changes in different parts of column test with and without plants (mg).

	Shoots	Roots (in Topsoil)	Roots (in Subsoil)	Topsoil Reduction	Subsoil Increase	Leaching Out
Planted test 1	0.0640	0.0567	0.0138	0.3626	0.0496	0.1785
Planted test 2	0.1499	0.0642	0.0116	0.5616	0.0960	0.2399
No plant test 1	-	-	-	0.0418	0.0209	0.0209
No plant test 2	-	-	-	0.1677	0.0207	0.1470

**Table 10 plants-12-02321-t010:** Chemical properties and Cd contents of the tested soil.

	Column Test 1		Column Test 2		Lysimeter Test	
	Topsoil	Subsoil	Topsoil	Subsoil	Topsoil	Subsoil
pH	5.57 ± 0.09	4.47 ± 0.04	5.27 ± 0.01	6.27 ± 0.04	4.95 ± 0.05	4.86 ± 0.11
OM g·kg^−1^	34.42 ± 0.08	5.60 ± 0.02	39.94 ± 0.17	10.51 ± 0.30	39.53 ± 4.43	16.10 ± 2.46
TN g·kg^−1^	1.96 ± 0.10	0.29 ± 0.02	1.27 ± 0.12	0.62 ± 0.06	2.24 ± 0.03	0.89 ± 0.11
TP g·kg^−1^	0.75 ± 0.08	0.26 ± 0.02	0.44 ± 0.02	0.16 ± 0.02	0.47 ± 0.04	0.24 ± 0.02
TK g·kg^−1^	7.59 ± 0.19	3.29 ± 0.62	9.56 ± 0.15	8.34 ± 0.30	10.43 ± 0.13	12.63 ± 0.20
Cd mg·kg^−1^	0.7676 ± 0.0077	0.0181 ± 0.0019	1.3058 ± 0.0232	0.0698 ± 0.0078	1.0236 ± 0.0294	0.1472 ± 0.0298

Note: OM, organic matter; TN, total nitrogen; TP, total phosphorus; TK, total potassium.

## Data Availability

The data that support the findings are presented in this paper. Other data are available from the corresponding author upon reasonable request.

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
