# Peer review of "Phytoextraction and Migration Patterns of Cadmium in Contaminated Soils by *Pennisetum hybridum"

_plants, 2023, doi:10.3390/plants12122321_

Round 1
Reviewer 1 Report
The study assesses the phytoremediation potential of an energy plant Pennisetum hybridum for remediation of cadmium. The work has been meticulously carried out and has potential for remediating cadmium contaminated environments. However, the manuscript has less of new references, in fact only 7 of the last 5 years. I suggest incorporation of some recent references that will make the manuscript more updated in terms of review of literature. So, I recommend minor revision of the article after incorporating the above-mentioned changes.
Author Response
Dear reviewer,
Thank you very much for your valuable comments and suggestions.
According your suggestion, we changed 10 references and most of them are new references of the last 5 years.
The revised version is attached. The modifications are done with tracked changes to highlight the revision (The added text is in red fond).
Thank you again for your kind attention to our manuscript!
Yours sincerely,
Qi-Tang Wu
The corresponding author

Reviewer 2 Report
Title: Phytoextraction and Migration Patterns of Cadmium in Contaminated Soils by Pennisetum hybridum
The authors studied and reported on Phytoextraction and Migration Patterns of Cadmium in Contaminated Soils by Pennisetum hybridum. The methodology was well organized and the results were properly discussed. The work was suitable for the publication. However, before going to accept, few corrections need to be incorporated.
1. Abstract should write more precisely
2. Each parts of the manuscript needs to be improved.
https://link.springer.com/chapter/10.1007/978-981-13-9117-0_1
https://www.nature.com/articles/s41598-017-05879-9
3. Discussion part has to be improved
4. Conclusions needs to revise more precisely.
5. Grammatical mistakes need to be fixed.
Title: Phytoextraction and Migration Patterns of Cadmium in Contaminated Soils by Pennisetum hybridum
The authors studied and reported on Phytoextraction and Migration Patterns of Cadmium in Contaminated Soils by Pennisetum hybridum. The methodology was well organized and the results were properly discussed. The work was suitable for the publication. However, before going to accept, few corrections need to be incorporated.
1. Abstract should write more precisely
2. Each parts of the manuscript needs to be improved.
https://link.springer.com/chapter/10.1007/978-981-13-9117-0_1
https://www.nature.com/articles/s41598-017-05879-9
3. Discussion part has to be improved
4. Conclusions needs to revise more precisely.
5. Grammatical mistakes need to be fixed.
Reviewer 3 Report
Some additional, specific comments:
1. What is the main question addressed by the research?
The phytoremedial potential of Pennisetum hybridum was studied, namely, the absorption of cadmium from soils
2. Do you consider the topic original or relevant in the field? Does it address a specific gap in the field?
The originality of the study is that the mechanism of absorption of cadmium by Pennisetum hybridum has been deeply studied, which makes it possible to manage the ecological state of agricultural lands through phytoremediation
3. What does it add to the subject area compared with other published material?
The subcellular distribution of Cd has been studied, which makes it possible to determine the location and shape of Cd in the roots of P. hybridum. It has been established that the binding of the root cell wall is the main mechanism that allows P. hybridum to retain heavy metals in its roots.
4. What specific improvements should the authors consider regarding the methodology? What further controls should be considered?
Methodology and controls are in order. However, in the discussion section it is desirable to see recommendations on the use of Pennisetum hybridum biomass after the plants have improved the soil (after their use as a Cd absorber)
5. Are the conclusions consistent with the evidence and arguments presented and do they address the main question posed?
Conclusions are logical and well founded
6. Please include any additional comments on the tables and figures.
It will be useful to add the technology of using these plants to improve the condition of the soil. You can present this in the form of a step-by-step diagram indicating the necessary actions.
